# Can the application of machine learning to electronic health records guide antibiotic prescribing decisions for suspected urinary tract infection in the Emergency Department?

Patrick Rockenschaub[1]*, Martin J. Gill[2], Dave McNulty[3], Orlagh Carroll[4], Nick Freemantle[5], Laura Shallcross[1]

1 Institute of Health Informatics, University College London, London, United Kingdom, 2 Department of Clinical Microbiology, Queen Elizabeth Hospital Birmingham, University Hospitals Birmingham NHS Foundation Trust, Birmingham, United Kingdom, 3 Health Informatics, University Hospitals Birmingham NHS Foundation Trust, Birmingham, United Kingdom, 4 Department of Medical Statistics, London School of Hygiene and Tropical Medicine, London, United Kingdom, 5 Institute of Clinical Trials and Methodology, University College London, London, United Kingdom

* patrick.rockenschaub.15@ucl.ac.uk

**Data Availability Statement:** The data that support the findings of this study are available from University Hospitals Birmingham NHS Foundation

## Abstract

Urinary tract infections (UTIs) are a major cause of emergency hospital admissions, but it remains challenging to diagnose them reliably. Application of machine learning (ML) to routine patient data could support clinical decision-making. We developed a ML model predicting bacteriuria in the ED and evaluated its performance in key patient groups to determine scope for its future use to improve UTI diagnosis and thus guide antibiotic prescribing decisions in clinical practice. We used retrospective electronic health records from a large UK hospital (2011–2019). Non-pregnant adults who attended the ED and had a urine sample cultured were eligible for inclusion. The primary outcome was predominant bacterial growth $\geq 10^4$ cfu/mL in urine. Predictors included demography, medical history, ED diagnoses, blood tests, and urine flow cytometry. Linear and tree-based models were trained via repeated cross-validation, re-calibrated, and validated on data from 2018/19. Changes in performance were investigated by age, sex, ethnicity, and suspected ED diagnosis, and compared to clinical judgement. Among 12,680 included samples, 4,677 (36.9%) showed bacterial growth. Relying primarily on flow cytometry parameters, our best model achieved an area under the ROC curve (AUC) of 0.813 (95% CI 0.792–0.834) in the test data, and achieved both higher sensitivity and specificity compared to proxies of clinician's judgement. Performance remained stable for white and non-white patients but was lower during a period of laboratory procedure change in 2015, in patients $\geq 65$ years (AUC 0.783, 95% CI 0.752–0.815), and in men (AUC 0.758, 95% CI 0.717–0.798). Performance was also slightly reduced in patients with recorded suspicion of UTI (AUC 0.797, 95% CI 0.765–0.828). Our results suggest scope for use of ML to inform antibiotic prescribing decisions by improving diagnosis of suspected UTI in the ED, but performance varied with patient characteristics. Clinical utility of predictive models for UTI is therefore likely to differ for important patient subgroups including women <65 years, women $\geq 65$ years, and men. Tailored models and

Trust, but restrictions apply to the availability of these data to protect individual confidentiality; they are not publicly available. Data are however available from Suzy Gallier (Head of Informatics Research & Commercial Development at University Hospitals Birmingham NHS Foundation Trust; email: staar@uhb.nhs.uk) upon reasonable request and with permission of University Hospitals Birmingham NHS Foundation Trust. The code used for all model development and evaluation in this study can be found at https://github.com/prockenschaub/uti-prediction.

**Funding:** This work was supported by a National Institute for Health Research (NIHR) Clinician Scientist award CS-2016-16-007 and the Rosetrees & the Stoneygate Trusts M627 (both to LS). The views expressed are those of the authors and not necessarily those of the NIHR or the Department of Health and Social Care. The funders had no role in study design, data collection and analysis, decision to publish, or preparation of the manuscript.

**Competing interests:** The authors have declared that no competing interests exist.

decision thresholds may be required that account for differences in achievable performance, background incidence, and risks of infectious complications in these groups.

## Author summary

Urinary tract infections (UTIs) often lead to emergency hospital visits, but they can be difficult to diagnose. As a result, antibiotics are often prescribed inappropriately. We created a machine learning model to help doctors better diagnose UTIs and prescribe antibiotics only when needed. We used health records from a large UK hospital and considered factors such as patient age, sex, ethnicity, urinary symptoms, laboratory tests, and medical history. Our model was good at predicting UTIs and performed better than doctors' guesses in many cases. It worked well for both white patients and ethnic minorities, but there were some differences in how well it did for older people, men, and patients who already had a suspected UTI. In summary, our study suggests that machine learning can help improve UTI diagnosis and antibiotic prescribing decisions in the emergency department. However, we might need to customize the model for different patient groups, as its performance varied based on patient characteristics.

## Background

Urinary tract infections (UTIs) are a major cause of emergency admissions in high-income countries [1,2] with annual costs estimated in excess of $2.8 billion in the US alone [2]. However, the ability to diagnose UTI reliably in the emergency department (ED) and differentiate it from other reasons for attendance is undermined by a lack of rapid and accurate diagnostic tests for UTI [3], the fact that patients often present with non-specific symptoms (particularly in the elderly) [4], and the need to make rapid diagnostic decisions. Previous studies have therefore repeatedly reported both over- and undertreatment of suspected UTI in the ED [5,6].

Recently, researchers have started investigating whether the application of risk models to data that are routinely collected during ED visits may be used to support earlier diagnosis of UTI and guide antibiotic initiation [7–10]. In the largest study to date, Taylor *et al.* [7] showed that machine learning can predict bacteriuria with high accuracy using data from 80,000 ED patients who presented with symptoms that were broadly compatible with suspected UTI. Their model achieved both higher sensitivity and specificity when compared to retrospective proxies of clinicians' judgement. Similar results were reported by Müller *et al.* [8] on a smaller Swiss cohort. However, average performance measures alone may be insufficient to judge the utility of these models in clinical practice.

Due to the need for large sample sizes, previous models were developed using data from heterogeneous patient groups. Many patients included in these studies are actually at very low risk of UTI, attending the ED for other reasons—including non-specific symptoms like altered mental status, other infections such as pneumonia, or even non-infectious conditions like heart disease—and receiving routine investigations for UTI [6]. This makes it difficult to determine their value in the primary target population of patients with suspected UTI. Successful deployment of predictive models for UTI requires good performance in this more narrowly defined target population, and may further need to distinguish between clinically important subgroups such as younger women (<65 years), older women (≥65 years), and men. These

groups differ in their background incidence of UTI, prevalence of asymptomatic bacteriuria (which does not usually require treatment), and risk of complications [11], and model performance and interpretation may vary as a result. Finally, predictive models need to show that they can achieve satisfactory performance without re-enforcing existing healthcare inequalities originating for example from race or ethnicity ("fair AI") [12].

In this study, we built on the work of Taylor et al. [7] to develop a model to predict bacteriuria in samples obtained from patients attending the ED in a large English hospital, which we evaluated in a temporally independent dataset. To explore scope to deploy such a model in clinical practice, we evaluated its performance in key patient subgroups including: age (<65 and ≥65 years), gender, ethnicity (white, non-white), and UTI syndrome at presentation (urinary symptoms, lower UTI, pyelonephritis, urosepsis).

## Methods

### Data and study population

We used electronic health record (EHR) data collected routinely in the ED at Queen Elizabeth Hospital Birmingham (QEHB), which serves an ethnically diverse population in southwest Birmingham. Approximately 115,000 ED patients attend QEHB each year. A detailed explanation of the study data was published previously [13]. In short, we included all adult patients who attended the ED at QEHB between November 1st 2011 and March 31st 2019 and who had a urine sample sent for microbiological testing within 24 hours of arrival (S1 Fig). After being seen by the ED physician, each patient at QEHB is routinely assigned one or more symptoms and/or suspected diagnoses out of a predefined list of ~800 ED diagnostic codes (e.g., pyelonephritis, haematuria, or acute confusion; S1 Text) [14]. We excluded patients without a valid record of age or sex, patients aged <18 years, pregnant women (identified via a pregnancy-related discharge diagnosis within ±9 months of arrival), and those who had a urine sample that was not ultimately cultured. As our focus was community-onset UTI, we excluded patients whose earliest urine sample was taken more than 24 hours after their recorded arrival in the ED (to account for delays in delivering samples to the laboratory but exclude hospital-acquired infection), and those with a previous diagnosis of UTI recorded ≤30 days before the date of ED attendance.

### Ethics approval and consent to participate

This research study was deemed exempt from NHS Research Ethics Committee review as there is no change to treatment or services or any study randomisation of patients into different treatment groups, and the study uses de-identified routinely collected data. Approval to undertake the study was obtained from the UK Health Research Authority ref: 17/HRA/3427.

### Outcome

The binary primary outcome was bacterial growth in the ED urine sample defined as growth of a predominant pathogen $\geq 10^4$ colony-forming units per millilitre (cfu/mL) during microbiological culture. Growth of several different organisms at once was considered mixed growth (unless growth of *Escherichia coli* was explicitly recorded) and—following standard procedure at QEHB—was considered sample contamination [15]. In order to not miss bloodstream infections from a urinary source, urine samples were also considered positive if they showed bacterial growth $<10^4$ cfu/mL but the same pathogen was also grown from a blood sample taken within 24 hours of arrival.

Notably, whether sent urines were actually cultured in the laboratory depended on the number of bacteria and white blood cells (WBCs) estimated from urine flow cytometry. The threshold values for proceeding to culture were WBC > 40/μL or bacteria > 4000/μL before October 2015 and were increased to WBC > 80/μL or bacteria > 8000/μL thereafter [13].

## Candidate predictors

Candidate predictors were selected based on clinical expertise, previous literature, and availability of data within the EHR system [13]. Considered information included age at arrival (in ten-year age-bands), sex, ethnicity (Asian, Black, White, Other), Charlson Comorbidity Index (CCI), presence of underlying renal/urological conditions, previous hospital or emergency visits for UTI or other reasons, blood tests (WBC, platelets, C-reactive protein [CRP], creatinine, alkaline phosphatase [ALP], bilirubin), urine flow cytometry (bacteria, WBC, red blood cells [RBC], epithelial cells, casts, crystals), calendar time (month, day of year, day of week, time of day). Suspected ED diagnosis was also included and grouped into UTI syndromes (lower UTI, pyelonephritis, urosepsis), UTI symptoms (urinary symptoms, abdominal pain, altered mental status), other infections (LRTI, sepsis of other origin, other infections), or non-infectious. A detailed list of the definition of each variable can be found in Rockenschaub *et al.* [13]. If more than one value was recorded for a variable during a patient's time in the ED, the mean value was included. Immunosuppression, vital signs, and previous antibiotic-resistant urine organisms were excluded from the analysis due to them being recorded in <10% of patients. Following Taylor *et al.* [7], we also considered a reduced set of predictors—which could be more easily implemented as a model in the ED—using only age, sex, history of positive urine culture, and all available urine flow cytometry measurements.

## Statistical analysis

**Patient characteristics.** Predictors were summarised for all patients and for those who did/didn't have a positive urine culture. Continuous variables were described using mean and standard deviation (if approximately normally distributed) or median and interquartile range (if non-normal). Categorical variables were described via counts and percentages. Differences in variable distribution by culture status were tested via t-test (normal continuous variables), Wilcoxon rank-sum test (non-normal continuous variables), and $\chi^2$ test (categorical variables).

**Predictive modelling.** For the predictive modelling, continuous predictors were capped at the 1st and 99th percentile ("winsorised") and transformed to approximate normality using Yeo-Johnson transformations. Since we observed at least some missingness for most of our variables, we considered three increasingly complex imputation strategies: mean imputation, k-nearest neighbour imputation, and multivariate imputation by chained equations (see S2 Text for a detailed description of each).

We considered the following predictive algorithms: standard logistic regression (LR), logistic regression with fractional polynomials (LR-FP), elastic net (E-NET), random forest (RF), and extreme gradient boosting (XGB). For LR-FP, up to four degrees of freedom (equivalent to two polynomial terms) were considered and the best fitting one chosen via the Akaike Information Criterion (AIC) [16]. For E-NET, RF, and XGB, 30 hyperparameter combinations were randomly chosen (see S1 Table) [17] and the best performing combination was chosen after internal validation.

All models were trained on data up to December 2017. Data from January to March 2018 were set aside for recalibration, and data from April 2018 to March 2019 were reserved as a temporally external test set. Training and internal validation was performed on the training data via 10-times repeated 10-fold cross validation, with all transformations and imputations being performed separately for each run to avoid data leakage. The best model of each algorithm class

was externally validated on the test set (with and without recalibration using Platt scaling). Discriminative performance was evaluated using the area under the receiver operating characteristic (AUC), specificity, and negative predictive value (NPV). Thresholds for the calculation of specificity and NPV were chosen such that a predefined sensitivity of 95% was achieved [13]. Difference in performance between models was tested via resampling (S3 Text). Calibration was assessed using calibration plots with locally estimated scatterplot smoothing.

**Sensitivity analyses.** In addition to the main analysis, we performed a broad set of sensitivity analyses to assess the robustness of our best model in specific situations and patient subgroups and to determine if there may be scope to deploy the model in clinical practice. We investigated changes in performance over time by re-running all analyses only on data before 2013 and testing it on data from 2013. We repeated this process for the years 2014, 2015, etc. Next, we evaluated the performance by age (<65 and ≥65 years), sex, ethnicity (white, non-white), and in the subgroup of patients with recorded suspicion of UTI indicated by an ED diagnosis of urinary symptoms, lower UTI, pyelonephritis, or urosepsis. The effect of mixed culture growth on our results was assessed by considering it as a positive culture or by excluding it from the analysis altogether. Finally, performance of our model was compared to two previously used proxies of clinicians' judgement [7]: ED diagnosis of UTI (lower UTI, pyelonephritis, or urosepsis) and/or prescription of systemic antibiotics recommended for UTI in QEHB's 2018 prescribing guidelines (see supporting information).

All analysis was performed in R (v3.6.2) and RStudio (v1.2.5033) on Windows 10. A prospective protocol for this analysis was published in Rockenschaub et al. [13]. All results were reported following the strengthening the Transparent Reporting of a multivariable prediction model for Individual Prognosis Or Diagnosis (TRIPOD) statement (S1 Checklist) [18].

## Results

Between 2011 and 2019, 795,752 ED visits were recorded at QEHB (S1 Fig). Of those, 23,128 (2.9%) visits from 18,353 unique patients had a urine sample submitted for microbiological analysis. After applying exclusion criteria, we assigned 10,352 (81.6%) visits to the training set, 479 (3.8%) visits to the calibration set, and 1,538 (12.1%) visits to the test set. Among included visits, 33% resulted in a positive urine culture which notably increased to 44% after thresholds for culture were raised in 2015 (see Methods and S2 Fig).

### Patient characteristics

Half (51.9%) of included visits were from patients ≥65 years and two-thirds (66.0%) were from women (Table 1). 23.8% and 17.9% of patients had CCIs of 1–2 and ≥3 respectively. History of renal (21.6%) or urological (28.5%) disease were common. Many included patients also had a hospital visit (47.8%) and/or urine sample (48.9%) recorded in the previous year. Over a third (39.4%) of included visits had a recorded ED diagnosis of UTI, with another 5.1% showing a record of urinary symptoms.

Bacterial growth was more commonly found among older patients, women, those of white ethnicity, and patients who previously had a positive urine culture (Table 1). It was also more commonly found among those with a recorded ED diagnosis of UTI (lower UTI, pyelonephritis, urosepsis) but not among those with only symptoms of UTI and was strongly associated with urine flow cytometry results and some blood tests, most notably CRP and platelet counts.

### Predictive modelling

The best performing model to predict bacteriuria was an XGB including all predictors, which achieved an AUC of 0.813 (95% CI 0.792–0.834; Table 2 and Fig 1) during external validation

**Table 1. Demography, medical history, and clinical characteristics at presentation in the ED.**

| | Overall | Bacterial growth | | Missing % | p-value |
|---|---|---|---|---|---|
| | | Yes | No | | |
| Number of visits | 12,680 (100.0) | 4,677 (36.9) | 8,003 (64.1) | | |
| **Demographics** | | | | | |
| ≥65 years (%) | 6,584 (51.9) | 2,645 (40.2) | 3,939 (59.8) | 0.0 | <0.001 |
| Female (%) | 8,368 (66.0) | 3,260 (39.0) | 5,108 (61.0) | 0.0 | <0.001 |
| Ethnicity (%) | | | | 5.3 | <0.001 |
| Asian | 1,671 (13.9) | 555 (33.2) | 1,116 (66.8) | | |
| Black | 552 (4.6) | 175 (31.7) | 377 (68.3) | | |
| White | 9,256 (77.1) | 3,496 (37.8) | 5,760 (62.2) | | |
| Other | 526 (4.4) | 183 (34.8) | 343 (65.2) | | |
| **Comorbidities** | | | | | |
| Charlson comorbidity index (%) | | | | 0.0 | 0.295 |
| 0 | 7,386 (58.2) | 2,686 (36.4) | 4,700 (63.6) | | |
| 1–2 | 3,023 (23.8) | 1,126 (37.2) | 1,897 (62.8) | | |
| ≥3 | 2,271 (17.9) | 865 (38.1) | 1,406 (61.9) | | |
| Cancer (%) | 915 (7.2) | 322 (35.2) | 593 (64.8) | 0.0 | 0.286 |
| Underlying renal condition (%) | 2,733 (21.6) | 1,011 (37.0) | 1,722 (63.0) | 0.0 | 0.913 |
| Underlying urological condition (%) | 3,614 (28.5) | 1,333 (36.9) | 2,281 (63.1) | 0.0 | 1.000 |
| Renal/urological surgery (%) | 2,484 (19.6) | 838 (33.7) | 1,646 (66.3) | 0.0 | <0.001 |
| **Hospital activity in prior year** | | | | | |
| Any hospitalisation (%) | 6,067 (47.8) | 2,259 (37.2) | 3,808 (62.8) | 0.0 | 0.446 |
| Urine sample taken (%) | 6,195 (48.9) | 2,251 (36.3) | 3,944 (63.7) | 0.0 | 0.217 |
| Urine sample positive (%) | 3,062 (24.1) | 1,355 (44.3) | 1707 (55.7) | 0.0 | <0.001 |
| Antibiotics in hospital (%) | 3,194 (25.2) | 1,149 (36.0) | 2,045 (64.0) | 0.0 | 0.225 |
| **Presentation in the ED** | | | | | |
| Recorded ED diagnosis (%) | | | | 6.7 | <0.001 |
| UTI | 4,998 (42.2) | 2,290 (45.8) | 2,708 (54.2) | | |
| UTI symptoms | 1,737 (14.7) | 476 (27.4) | 1,261 (72.6) | | |
| Other infection | 1,684 (14.2) | 495 (29.4) | 1,189 (70.6) | | |
| Other diagnoses | 3,414 (28.9) | 1,149 (33.7) | 2,265 (66.3) | | |
| Urine flow cytometry (median / IQR) | | | | | |
| Bacteria x$10^3$/μL | 8.1 (2.3, 21.3) | 13.5 (5.2, 33.3) | 5.5 (1.0, 14.3) | 15.4 | <0.001 |
| White blood cells x1/μL | 328 (111, 1191) | 540 (163, 1881) | 249 (96, 821) | 15.4 | <0.001 |
| Red blood cells x1/μL | 35.0 (13.0, 132.0) | 32.0 (12.0, 104.0) | 37.0 (14.0, 158.0) | 15.4 | <0.001 |
| Epithelial cells x1/μL | 21.0 (6.0, 54.0) | 14.0 (5.0, 40.0) | 25.0 (9.0, 62.0) | 15.4 | <0.001 |
| Small round cells x1/μL | 1.0 (0.0, 4.0) | 2.0 (1.0, 5.0) | 1.0 (0.0, 3.0) | 15.4 | <0.001 |
| Casts x1/μL | 1.0 (0.0, 2.0) | 1.0 (0.0, 3.0) | 1.0 (0.0, 2.0) | 15.4 | <0.001 |
| Crystals x1/μL | 5.0 (2.0, 15.0) | 5.0 (2.0, 16.0) | 5.0 (2.0, 13.0) | 60.0 | 0.004 |
| Blood tests (median / IQR) | | | | | |
| C-reactive protein mg/L | 27 (6, 93) | 32 (7, 98) | 24 (6, 89) | 53.9 | <0.001 |
| White blood cells x$10^3$/μL | 10.8 (8.0, 14.5) | 10.9 (8.2, 14.6) | 10.7 (7.9, 14.5) | 43.8 | 0.088 |
| Platelets x$10^3$/μL | 231 (182, 292) | 227 (180, 284) | 233 (183, 296) | 44.0 | 0.001 |
| Creatinine μmol/L | 84 (66, 118) | 83 (65, 115) | 84 (66, 120) | 45.1 | 0.045 |
| Bilirubin μmol/L | 9 (6, 14) | 9 (6, 14) | 9 (6, 14) | 51.4 | 0.140 |

*(Continued)*

**Table 1.** (Continued)

| | Overall | Bacterial growth | | Missing % | p-value |
|---|---|---|---|---|---|
| Alkaline phosphatase IU/L | 88 (69, 117) | 88 (69, 117) | 88 (69, 119) | 51.4 | 0.897 |

p-values refer to the result of a t-test for difference in means (normal continuous variables), a Wilcoxon rank-sum test for difference in means (non-normal continuous variables), or a χ2 test of independence (categorical variables).

ED, emergency department; IQR, interquartile range.

on temporally independent data. At a pre-defined sensitivity of 95%, the model achieved a specificity of 36.1% (95% CI 30.9–40.9) and had an NPV of 89.8% (95% CI 87.9–91.3). Both a simple LR using all predictors (AUC 0.796, 95% CI 0.776–0.817) and an XGB using the reduced set of predictors (AUC 0.806, 95% CI 0.783–0.828) performed comparably, although p-values from bootstrapping suggested slightly lower performance of LR compared to a full XGB (p<0.001). Results from internal validation were similar but performance was slightly worse using multiple imputation (S2 Table and S3 Table). The primary importance of urine flow cytometry results—which make up most predictors in the reduced set—for discriminative power was also observed in univariate analyses (S4 Table). The final XGB model tended to underestimate the risk of bacterial growth, which was (over-)corrected after re-calibration (Fig 2).

## Sensitivity analyses

Coinciding with changes in laboratory procedures, estimated performance of our XGB model was reduced around 2015 (AUC 0.766, 95% CI 0.740–0.793; Fig 3). Reduced performance was also seen in patients aged ≥65 years (AUC 0.783, 95% CI 0.752–0.815) and in men (AUC 0.758, 95% CI 0.717–0.798) with bootstrapped p-values for a difference in performance of p = 0.004 and p<0.001 compared to those aged <65 years and compared to women. There was no significant difference in performance for patients with an ED diagnosis of lower UTI, pyelonephritis, urosepsis, or UTI symptoms (AUC 0.797, 95% CI 0.765–0.828, p = 0.210; Table 3), and no evidence that performance varied by ethnicity (AUC 0.831, 95% CI 0.780–

**Table 2. Discriminative performance of candidate models when predicting bacterial growth in 1,538 urine samples of a temporally external test set from 2018/19.**

| Model | AUC (95% CI) | Specificity (95% CI) | NPV (95% CI) | p-value |
|---|---|---|---|---|
| All candidate predictors | | | | |
| XGB | 0.813 (0.792–0.834) | 36.1 (30.9–40.9) | 89.8 (87.9–91.3) | - |
| RF | 0.803 (0.783–0.825) | 33.0 (26.1–39.0) | 88.8 (86.2–90.7) | 0.048 |
| LR | 0.796 (0.776–0.817) | 28.9 (23.5–35.5) | 87.6 (84.9–90.1) | <0.001 |
| E-NET | 0.796 (0.775–0.819) | 28.4 (23.3–36.0) | 87.4 (84.4–90.1) | <0.001 |
| LR-FP | 0.791 (0.771–0.813) | 32.8 (27.1–38.4) | 88.9 (86.6–90.8) | <0.001 |
| Reduced set of predictors | | | | |
| XGB | 0.806 (0.783–0.828) | 34.6 (29.8–39.4) | 89.4 (87.7–91.1) | 0.102 |
| E-NET | 0.787 (0.766–0.810) | 31.1 (24.9–36.0) | 86.5 (83.1–89.2) | <0.001 |
| LR | 0.786 (0.765–0.809) | 29.2 (24.2–34.3) | 85.9 (82.1–89.1) | <0.001 |
| LR-FP | 0.783 (0.762–0.807) | 33.9 (27.5–40.2) | 88.5 (85.9–90.5) | <0.001 |
| RF | 0.761 (0.736–0.785) | 26.7 (23.6–29.8) | 71.7 (66.3–76.8) | <0.001 |

Specificity and NPV were calculated at a predefined sensitivity of 95%. Confidence intervals were obtained via 1,000 bootstraps of the external test set. p-values indicate a difference in AUC compared to the top performing model and were calculated as the proportion of bootstraps in which the top performing model retained better performance [28,29], multiplied by two to account for the two-sided nature of our hypothesis.

AUC, area under the receiver operating characteristic; CI, confidence interval; E-NET, elastic net; LR, logistic regression; LR-FP, logistic regression with fractional polynomials; NPV, negative predictive value; RF, random forest; XGB, extreme gradient boosting trees.

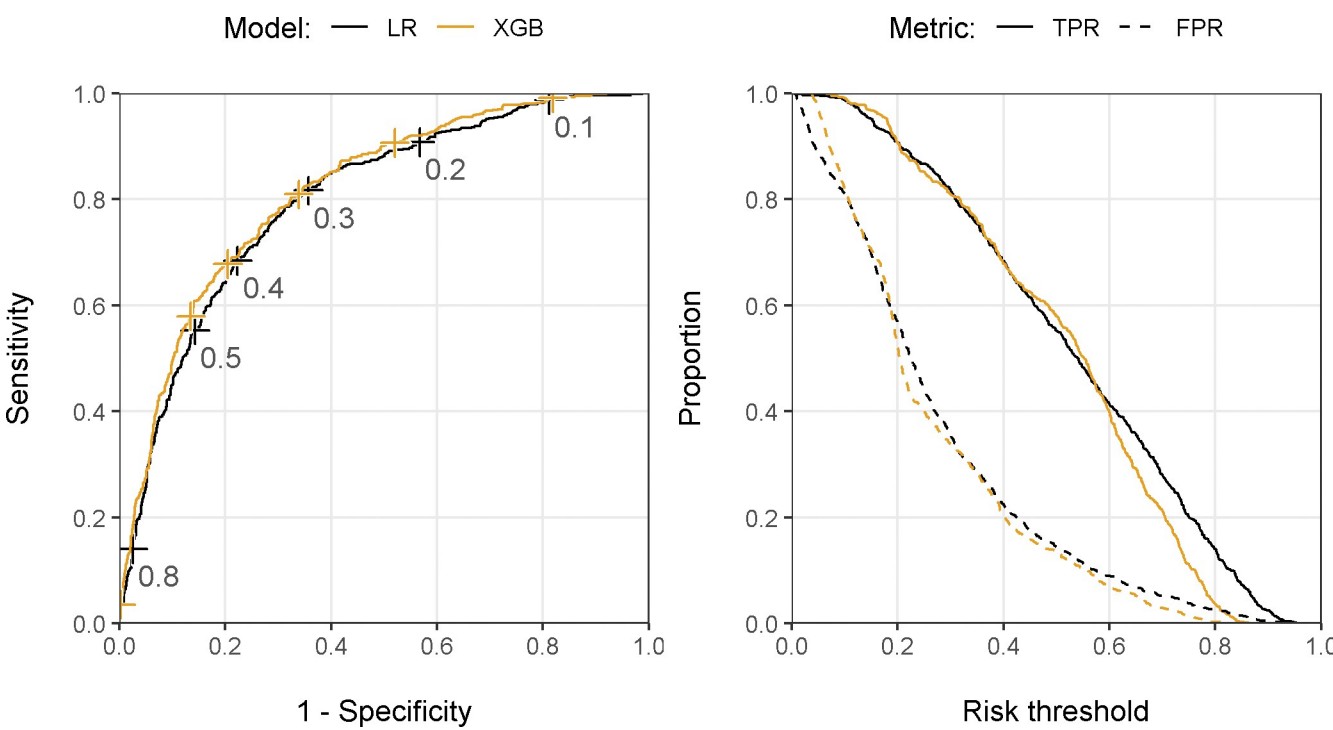

**Fig 1.** Receiver operating characteristic curve with risk thresholds (left panel) and classification plots (right panel) of the final LR and XGB models in the temporally external test set using all candidate predictors. FPR, false positive rate; LR, logistic regression; TPR, true positive rate; XGB, extreme gradient boosting trees.

0.873, p = 0.153). The model showed some miscalibration in subgroups, primarily in the elderly (Fig 4).

Estimated performance differed strongly depending on how the microbiological culture finding of mixed growth (23.5% of all samples) was classified, which is often considered indicative of a contaminated / unreliable sample [7,15]. When mixed culture growth was considered positive growth during training and testing, estimated external model performance increased to AUC 0.864 (95% CI 0.847–0.880), which further increased to AUC 0.892 (95% CI 0.875–0.909) if samples with mixed growth were excluded from the analysis altogether. Bootstrapping showed a clear difference in performance in both cases (p<0.001). Importantly, samples with mixed growth were frequently assigned high probabilities of bacteriuria, irrespective of how mixed growth was classified in the model (S3 Fig).

When compared to retrospective proxies of clinician's judgement (ED diagnosis of UTI and/or prescription of systemic antibiotics recommended for UTI), our model achieved both higher sensitivity and specificity (Table 4). At a specificity of 63.7%—which would be achieved by a model that predicts bacteriuria whenever there was a recorded ED diagnosis of UTI—our model obtained considerably higher sensitivity (83.0%, 95% CI 80.3–85.0 versus 48.2%, 95% CI 44.3–52.1). Conversely, at a sensitivity of 59.9% achieved by using recorded ED diagnosis of UTI and/or antibiotic prescribing to infer clinical judgement, our model achieved notably higher specificity (85.5%, 95% CI 83.1–87.7 versus 51.6%, 95% CI 48.3–55.0).

## Discussion

In this retrospective EHR study, our best-performing model was able to predict bacterial growth in ED urine samples with an AUC of 0.815 in an ethnically diverse patient population

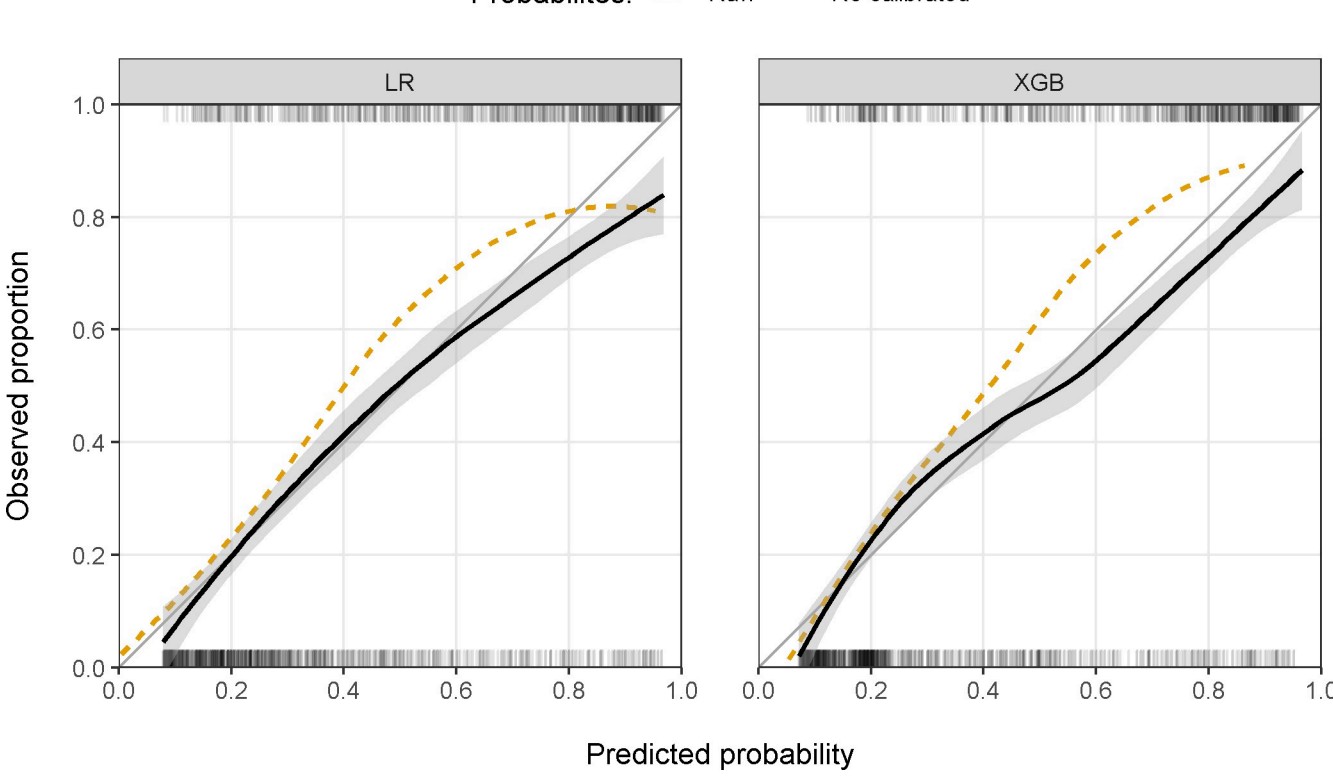

**Fig 2.** Model calibration of the final LR (left panel) and XGB (right panel) model before and after re-calibration in the temporally external test set using all candidate predictors. Calibration was estimated using LOESS regression and grey areas represent 95% confidence intervals. Ticks at the top (bacterial growth) and bottom (no bacterial growth) of each panel show the distribution of positive and negative samples in the test set. LOESS, locally estimated scatterplot smoothing; LR, logistic regression; XGB, extreme gradient boosting trees.

and outperformed retrospective proxies of clinical judgement. However, performance differed over time and depending on the patient population in which it was used, with reduced performance in patients aged ≥65 years and men. Given the differences in UTI incidence, prevalence of asymptomatic bacteriuria, and risk of infectious complications in these important target populations, this has implications for the model's potential use to predict UTI and thus guide antibiotic prescribing in clinical practice and may suggest the need for separate models or thresholds for decision making.

Our model primarily relied on urine flow cytometry parameters when making its predictions. A reduced model based on age, sex, history of positive urine culture, and urine flow cytometry performed almost as well as a model using all predictors. Some flow cytometry results—bacterial count and WBC—were already used at QEHB's laboratory during the study period in a simple decision rule to screen for samples with extremely low probability of bacterial growth, which were then excluded from culture. The good predictive power of our model even in pre-screened urines suggests that the value of flow cytometry to support early diagnosis of bacteriuria and UTI may currently be underused in clinical practice. For example, if the model were used, it would have correctly identified 95% of samples which later showed bacterial growth while ruling out bacterial growth early in 36.6% of ultimately culture-negative samples. Of those samples that were flagged as likely negative, 90.3% were correctly classified and did not exhibit bacterial growth during culture. This highlights the potential of data-driven models to aid the diagnosis of UTI and to reduce laboratory cost if clinical parameters are used in addition to cytometry to select which urines are cultured.

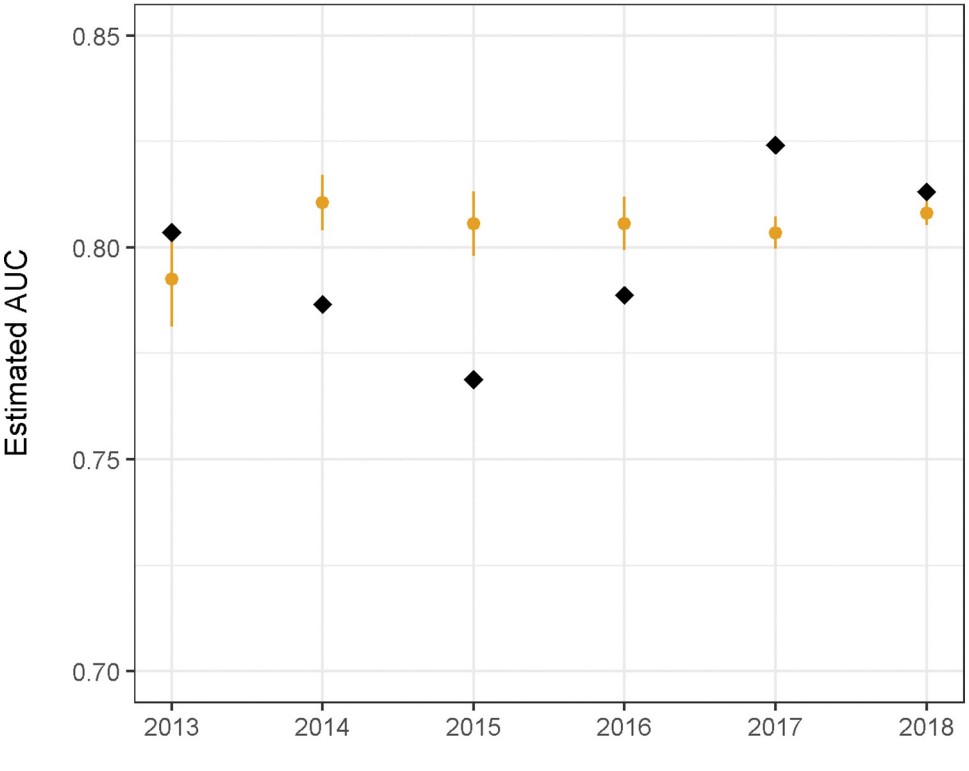

**Fig 3. Changes in the estimated AUC of the best-performing model (XGB using all predictors) over time when predicting bacterial growth.** For each year, data up to that year were used to train the model. Orange dots and lines represent estimated performance from internal validation of the training data before that year. Black diamonds represent the estimated performance from external validation using data from that year.

While our model thus achieved good performance, this was lower than previously reported results from both the US (AUC 0.904) [7] and Switzerland (AUC 0.930) [8]. Reasons for this discrepancy are not immediately obvious, but important differences exist with regards to the data used for training and evaluation. Whereas only 2.9% of ED patients in our study had a urine culture requested, 25.6% of ED patients in the US-based study by Taylor *et al.* had a culture requested [7]. Although propensity to culture might genuinely be higher in the US, nation-wide estimates suggest much lower rates of 8.1% [19] and another US single centre study reported rates as low as 2.3% [20]. The US study may therefore have been subject to selection bias, or—if urine cultures were indeed requested for one in four patients attending the ED—was not representative of other hospitals in the UK and US. Patient denominators are not available for the Swiss study by Müller *et al.* [8]. However, Müller *et al.* treated mixed growth as positive growth, which was also associated with higher performance in our study. Furthermore, samples that were *a priori* dismissed by our laboratory due to low bacteria or urinary WBC counts *were* cultured in Switzerland. These samples were unlikely to grow bacteria (S3 Fig), thus representing "easy wins". As a result, the algorithm developed by the Swiss authors would be expected to perform worse when transferred to our patient population and

**Table 3. Discriminative performance of our best-performing XGB model in clinically relevant patient subpopulations.**

| Subgroup | AUC (95% CI) | Specificity (95% CI) | NPV (95% CI) | p-value |
|---|---|---|---|---|
| Age | | | | 0.004 |
| <65 years | 0.842 (0.813–0.870) | 36.8 (31.1–44.9) | 91.8 (88.2–95.1) | |
| ≥65 years | 0.783 (0.752–0.815) | 32.9 (28.3–38.8) | 87.5 (83.0–91.0) | |
| Sex | | | | <0.001 |
| Male | 0.758 (0.717–0.798) | 30.9 (24.2–32.4) | 82.5 (75.9–87.4) | |
| Female | 0.840 (0.815–0.864) | 37.2 (32.6–43.6) | 93.7 (90.6–96.5) | |
| Ethnicity | | | | 0.153 |
| White | 0.804 (0.777–0.828) | 35.4 (29.3–39.9) | 89.9 (86.8–92.2) | |
| Asian, Black, or Other* | 0.831 (0.780–0.873) | 42.1 (31.8–49.2) | 91.0 (83.5–95.9) | |
| ED diagnosis | | | | 0.210[†] |
| UTI (including symptoms) | 0.797 (0.765–0.826) | 27.9 (22.8–34.0) | 90.7 (85.3–94.8) | |
| UTI (excluding symptoms) | 0.806 (0.775–0.838) | 27.6 (22.4–33.2) | 92.5 (87.4–97.5) | |
| Other diagnoses | 0.824 (0.794–0.853) | 40.7 (35.2–49.3) | 89.4 (86.4–92.3) | |

Specificity and NPV were calculated at a predefined sensitivity of 95%. Confidence intervals were obtained via 1,000 bootstraps of the external test set. p-value test for a difference between AUC of the two subgroups and were calculated as the proportion of bootstraps in which the subgroup with overall smaller performance remained smaller, multiplied by two to account for the two-sided nature of our hypothesis.

\* Due to otherwise small patient numbers in these subsets, ethnic minorities were considered as a single group. Patients with unknown ethnicity were excluded from this comparison.

[†] Difference between UTI (including symptoms) versus Other diagnoses.

AUC, area under the receiver operating characteristic; CI, confidence interval; NPV, negative predictive value; XGB, extreme gradient boosting trees.

clinical and laboratory practice. This emphasises the importance of understanding variation in laboratory processes, which can have a major impact on the implementation of ML models in clinical practice. It is reassuring, though, that both models—like ours—predominantly relied on urinalysis parameters in their predictions, which agrees with findings from non-ED populations [3,9].

## Strengths and limitations

To the best of our knowledge, this is the first model predicting bacterial growth in ED urines from a UK patient population. A major strength of this analysis is the use of a large sample of high-quality EHR data from a major teaching hospital. QEHB's long history of electronic record keeping [21] allowed us to use records collected over multiple years, perform extensive sensitivity analyses, and assess likely future model performance.

However, the data used in this analysis were nevertheless recorded as part of routine care rather than specifically for research. Our data contained missing data, which needed to be addressed by imputation. Some key variables relevant to the diagnosis of UTI were completely absent from the EHR data for the duration of our study, including urine dipstick results and prior antibiotic prescribing outside of hospital. Dipsticks are commonly used to support the diagnosis of UTI [3,22] and prior antibiotic use may have prevented the growth of microorganisms during culture [6], potentially limiting the model's power to predict bacterial growth [23]. Furthermore, a substantial proportion of urine samples included in this analysis were submitted for culture in the absence of any recorded suspicion of UTI or weren't cultured despite suspicion of UTI. While this likely reflects real-world clinical practice [4], clinical guidelines suggest that bacteriuria should only guide treatment in the presence of clear symptoms [24]. Reducing unnecessary investigations in patients who are very unlikely to have UTI

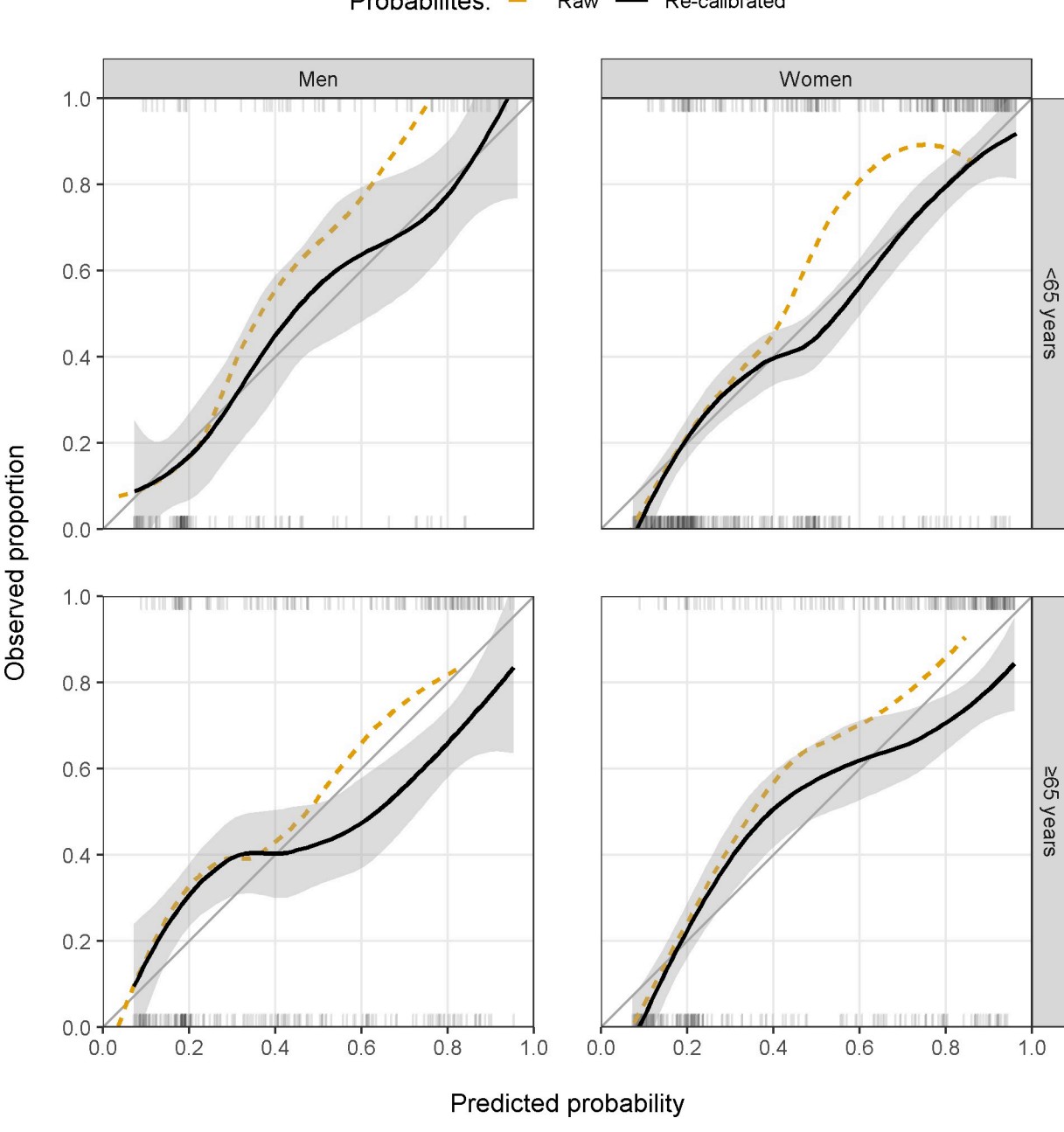

**Fig 4. Raw and re-calibrated model calibration of the final XGB model by age (<65 and ≥65 years) and sex (men and women).** Calibration was estimated using LOESS regression and gray areas represent 95% confidence intervals. Ticks at the top (bacterial growth) and bottom (no bacterial growth) of each panel show the distribution of positive and negative samples in the test set. LOESS, locally estimated scatterplot smoothing; XGB, extreme gradient boosting trees.

could bring cost savings and support antimicrobial stewardship. Further research is required to understand the reasons why these patients are being investigated for suspected UTI. Finally, definitions of clinical judgement in this study were inferred indirectly from retrospective data and do not reflect the full complexity of real-world clinical decision making, potentially underestimating clinicians' performance in predicting bacteriuria.

**Table 4. Comparison of discriminative model performance to proxies of clinical judgement as defined by a) ED diagnosis of UTI alone or b) ED diagnosis of UTI or prescription of systemic antibiotics in the ED.**

| | Accuracy (95% CI) | Sensitivity (95% CI) | Specificity (95% CI) |
|---|---|---|---|
| a) Proxy = UTI diagnosis (specific) | | | |
| Clinicians | 56.9 (54.3–59.3) | 48.2 (44.3–52.1) | 63.7 (60.5–66.7) |
| XGB | 72.3 (70.0–74.4) | 83.0 (80.3–85.9) | 63.8 (60.5–66.6)* |
| b) Proxy = UTI diagnosis AND/OR prescription of antibiotics indicated for UTI (sensitive) | | | |
| Clinicians | 55.3 (52.9–57.8) | 59.9 (53.6–63.5) | 51.6 (48.3–55.0) |
| XGB | 74.3 (72.0–76.4) | 60.0 (56.6–63.5)* | 85.5 (83.1–87.7) |

* These were matched to achieve the same performance as that estimated for clinicians

CI, confidence interval; ED, emergency department; UTI, urinary tract infection; XGB, extreme gradient boosting trees.

## Clinical, policy and research implications

Our results suggest a potential need for separate prediction models and decision thresholds in key populations such as the elderly or men. Variations in performance around a change in laboratory procedures in our observation period further demonstrate the difficulties of developing a single model that retains performance across time and hospitals. Instead of a one size fits all approach, it may be necessary to (re-)train and validate models using local data from the target population [25]. Fortunately, key variables such as urine flow cytometry remained stable across studies [7,8,22] and clinical settings [9], and there might be an opportunity to improve diagnosis of UTI simply by feeding back these raw results to clinicians in real-time.

Our results also highlight the prevalence of mixed growth in ED settings, with one quarter of cultured urine samples showing mixed growth. While generally regarded as sample contamination [7,15], some authors have argued that strict microbiological protocols might miss important bacteriuria [26,27]. Either way, mixed growth has important implications for models that aim to predict (predominant) growth and is difficult to predict with currently available diagnostics [8].

We suggest that our model may be embedded within the laboratory workflow. Assuming all relevant clinical data have been recorded at the time of urine sample submission and are readily available within the electronic patient record, the laboratory can use this data and our model to rate the results of flow cytometry and provide rapid feedback to ED clinicians. The results should be reported back to clinicians in such a way that it helps them decide on the likelihood of UTI and choice of antibiotic treatment. On-going education of clinical personnel and audit of the process will be necessary for a successful implementation.

## Conclusion

The ML models used in this study were able to predict bacterial growth in ED urine samples with good predictive accuracy but expected performance varied with patient characteristics. Effective deployment of predictive models to guide antibiotic prescribing decisions for UTI are likely to require tailored approaches for patient subgroups with a high prevalence of asymptomatic bacteriuria (patients aged ≥65 years) or high risk of complication (men).

## Supporting information

**S1 Checklist. Transparent Reporting of a multivariable prediction model for Individual Prognosis Or Diagnosis (TRIPOD) statement.**
(PDF)

**S1 Table. Hyperparameter ranges used for model tuning.**
(DOCX)

**S2 Table. Discriminative performance in internal validation.** AUC, Specificity, and NPV when predicting bacterial growth during 10-times repeated 10-fold cross-validation of the development set.
(DOCX)

**S3 Table. Discriminative performance by imputation method.** Estimated AUC of LR and XGB using all predictors during external validation, by imputation method.
(DOCX)

**S4 Table. Feature importance.** Top ten variables with the highest AUC when predicting bacterial growth during internal validation of LR and XGB.
(DOCX)

**S5 Table. Characteristics of train and test data.** Comparison of demography, medical history, and clinical characteristics between training set (before or in 2017) and test set (after 2017).
(DOCX)

**S6 Table. Parameters of final LR model.** Applied Yeo-Johnson transformations and final model coefficients for baseline LR models using all predictors or the reduced set of predictors.
(DOCX)

**S1 Fig. Flow chart of cohort selection for community-onset UTI in the ED at QEHB.**
(TIF)

**S2 Fig. Number of urine samples and proportion of culture-positive urines by year.** A) Yearly distribution of ED visits with a urine sample sent for microbiological culture (blue lines), and number of ED visits for which the urine sample was ultimately cultured (grey bars). Although visits before November 2011 are presented here to show an overall trend, they were not included in the main analysis since ED diagnoses were not yet recorded for these visits. B) Quarterly proportion of cultured urine samples that showed predominant bacterial growth (black dots) and linear trend (blue lines) before and after the change in urinalysis thresholds in October 2015.
(TIF)

**S3 Fig. Distribution of predictions by culture result.** Distribution of model predictions in the test set for samples with dominant growth, mixed growth, and no growth, depending on how mixed growth was treated during model training.
(TIF)

**S1 Text. Systematic antibiotics recommended for UTI.**
(DOCX)

**S2 Text. Methods for dealing with missing data.**
(DOCX)

**S3 Text. Testing differences in model performance.**
(DOCX)

## Author Contributions

**Conceptualization:** Patrick Rockenschaub, Nick Freemantle, Laura Shallcross.

**Data curation:** Patrick Rockenschaub, Dave McNulty.

**Formal analysis:** Patrick Rockenschaub.

**Funding acquisition:** Laura Shallcross.

**Investigation:** Patrick Rockenschaub, Laura Shallcross.

**Methodology:** Patrick Rockenschaub, Martin J. Gill, Orlagh Carroll.

**Supervision:** Martin J. Gill, Nick Freemantle, Laura Shallcross.

**Validation:** Patrick Rockenschaub.

**Visualization:** Patrick Rockenschaub.

**Writing – original draft:** Patrick Rockenschaub, Laura Shallcross.

**Writing – review & editing:** Patrick Rockenschaub, Martin J. Gill, Nick Freemantle, Laura Shallcross.

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
