## [Decision Letter · Decision Letter 0]

22 Mar 2023

PDIG-D-22-00301

Can the application of machine learning to electronic health records guide antibiotic prescribing decisions for suspected urinary tract infection in the Emergency Department?

PLOS Digital Health

Dear Dr. Rockenschaub,

Thank you for submitting your manuscript to PLOS Digital Health. After careful consideration, we feel that it has merit but does not fully meet PLOS Digital Health's publication criteria as it currently stands. Therefore, we invite you to submit a revised version of the manuscript that addresses the points raised during the review process.

Please submit your revised manuscript within 30 days. If you will need more time than this to complete your revisions, please reply to this message or contact the journal office at digitalhealth@plos.org. Please include the following items when submitting your revised manuscript:

We look forward to receiving your revised manuscript.

Kind regards,

Bo Wang

Academic Editor

PLOS Digital Health

Journal Requirements:

2. We ask that a manuscript source file is provided at Revision. Please upload your manuscript file as a .doc, .docx, .rtf or .tex.

3. Please provide separate figure files in .tif or .eps format only and remove any figures embedded in your manuscript file. Please also ensure that all files are under our size limit of 10MB.

4. We have noticed that you have uploaded Supporting Information files, but you have not included a list of legends. Please add a full list of legends for your Supporting Information files after the references list.

Additional Editor Comments (if provided):

Reviewers' comments:

Reviewer's Responses to Questions

**Comments to the Author**

1. Does this manuscript meet PLOS Digital Health’s publication criteria? Is the manuscript technically sound, and do the data support the conclusions? The manuscript must describe methodologically and ethically rigorous research with conclusions that are appropriately drawn based on the data presented.

Reviewer #1: Yes

Reviewer #2: Partly

2. Has the statistical analysis been performed appropriately and rigorously?

Reviewer #1: Yes

Reviewer #2: Yes

3. Have the authors made all data underlying the findings in their manuscript fully available (please refer to the Data Availability Statement at the start of the manuscript PDF file)?

Reviewer #1: No

Reviewer #2: No

4. Is the manuscript presented in an intelligible fashion and written in standard English?

Reviewer #1: Yes

Reviewer #2: Yes

5. Review Comments to the Author

Reviewer #1: In this paper, authors have applied some of well-known machine learning techniques for urinary tract infection predictions in emergency department from the retrospective electronic health records from a large UK hospital.

+ve points:

-this paper studies an important problem. 

-Overall, the paper is well written, clear and easy to follow.

- Sufficiently detailed and provides thorough empirical analysis of the results, including calibration and sensitivity analysis of the results

- Code is released publicly 

-ve points (I didn't find any major flaw with the paper but some minor issues/comments are given below):

- Use of proportions of bootstaps for the p-value is bit new to me. Could you provide some references/explanations.

- Authors mentiond different types of tests for checking the statistical significance. If you could elaborate (maybe in the supplemantry material) when and where were those used then that will be hepful for the readers.

- Authors mention that they used 30 random runs for hyperparameter tuning. Please also provide the name and range of values (maybe in the supplementary materials) used.

- What decision thresholds were used and how were they calculated?

Reviewer #2: The analysis in the article is clear and comprehensive. The description of the analysis and results has to be rewritten. There is no uniform way of presentation, some parts are very messy. The part of medical applications should be more detailed.

6. PLOS authors have the option to publish the peer review history of their article (what does this mean?). If published, this will include your full peer review and any attached files.

**Do you want your identity to be public for this peer review?** For information about this choice, including consent withdrawal, please see our Privacy Policy.

Reviewer #1: No

Reviewer #2: Yes: Anna Khalemsky

---

## [Decision Letter · Decision Letter 1]

25 Apr 2023

Can the application of machine learning to electronic health records guide antibiotic prescribing decisions for suspected urinary tract infection in the Emergency Department?

PDIG-D-22-00301R1

Dear Patrick Rockenschaub,

We are pleased to inform you that your manuscript 'Can the application of machine learning to electronic health records guide antibiotic prescribing decisions for suspected urinary tract infection in the Emergency Department?' has been provisionally accepted for publication in PLOS Digital Health.

Best regards,

Bo Wang

Academic Editor

PLOS Digital Health

Reviewer Comments (if any, and for reference):

Reviewer's Responses to Questions

**Comments to the Author**

1. If the authors have adequately addressed your comments raised in a previous round of review and you feel that this manuscript is now acceptable for publication, you may indicate that here to bypass the “Comments to the Author” section, enter your conflict of interest statement in the “Confidential to Editor” section, and submit your "Accept" recommendation.

Reviewer #1: All comments have been addressed

Reviewer #2: All comments have been addressed

2. Does this manuscript meet PLOS Digital Health’s publication criteria? Is the manuscript technically sound, and do the data support the conclusions? The manuscript must describe methodologically and ethically rigorous research with conclusions that are appropriately drawn based on the data presented.

Reviewer #1: Yes

Reviewer #2: Yes

3. Has the statistical analysis been performed appropriately and rigorously?

Reviewer #1: Yes

Reviewer #2: Yes

4. Have the authors made all data underlying the findings in their manuscript fully available (please refer to the Data Availability Statement at the start of the manuscript PDF file)?

Reviewer #1: Yes

Reviewer #2: Yes

5. Is the manuscript presented in an intelligible fashion and written in standard English?

Reviewer #1: Yes

Reviewer #2: Yes

6. Review Comments to the Author

Reviewer #1: Authors have addressed all my comments so I recommend the manuscript. Thank you!

Reviewer #2: I am satisfied with the changes the authors made in the paper.

7. PLOS authors have the option to publish the peer review history of their article (what does this mean?). If published, this will include your full peer review and any attached files.

**Do you want your identity to be public for this peer review?** For information about this choice, including consent withdrawal, please see our Privacy Policy.

Reviewer #1: **Yes: **Vinod Kumar Chauhan (Institute of Biomedical Engineering, University of Oxford UK)

Reviewer #2: No
